# HPV and Other Risk Factors Involved in Pharyngeal Neoplasm—Clinical and Morphopathological Correlations in the Southwestern Region of Romania

**DOI:** 10.3390/pathogens12080984

**Published:** 2023-07-27

**Authors:** Carmen Aurelia Mogoantă, Mircea Sorin Ciolofan, Anca-Maria Istrate-Ofițeru, Stelian-Ștefăniță Mogoantă, Gabriela-Camelia Roșu, Florin Anghelina, Alina-Nicoleta Căpitanescu, Ioana Cristina Opriscan, Nina Ionovici, Mihaela Roxana Mitroi, Oana Badea, Gheorghe Iovănescu

**Affiliations:** 1ENT Department, Faculty of Medicine, University of Medicine and Pharmacy of Craiova, 200349 Craiova, Romania; carmen_mogo@yahoo.com (C.A.M.); sorin.ciolofan@umfcv.ro (M.S.C.); anghelina.florin@gmail.com (F.A.); alinacapitanescu@yahoo.com (A.-N.C.); mhlmitroi@yahoo.com (M.R.M.); 2Histology Department, University of Medicine and Pharmacy of Craiova, 200349 Craiova, Romania; nicola_camelia92@yahoo.com; 3General Surgery Department, University of Medicine and Pharmacy of Craiova, 200349 Craiova, Romania; 4Doctoral School, University of Medicine and Pharmacy of Craiova, 200349 Craiova, Romania; ioana.opriscan@gmail.com; 5Department of Occupational Medicine, University of Medicine and Pharmacy of Craiova, 200349 Craiova, Romania; ninaionovici@yahoo.com; 6Department of Modern Languages, University of Medicine and Pharmacy of Craiova, 200349 Craiova, Romania; o_voiculescu@yahoo.com; 7ENT Department, Faculty of Medicine, University of Medicine and Pharmacy Victor Babes, 300041 Timisoara, Romania; giovanescu@gmail.com

**Keywords:** human papillomavirus, cell proliferation, tumor suppressor gene

## Abstract

Oropharyngeal squamous cell carcinoma (OPSCC) development is strongly associated with risk factors like smoking, chronic alcohol consumption, and the living environment, but also chronic human papilloma virus (HPV) infection, which can trigger cascade cellular changes leading to a neoplastic transformation. The prevalence of these factors differs among different world regions, and the prevention, diagnosis, and prognosis of OPSCC are highly dependent on them. We performed a retrospective study on 406 patients diagnosed with OPSCC in our region that were classified according to the tumor type, localization and diagnosis stage, demographic characteristics, risk factors, and histological and immunohistochemical features. We found that most of the patients were men from urban areas with a smoking habit, while most of the women in our study were diagnosed with tonsillar OPSCC and had a history of chronic alcoholism. During the immunohistochemical study, we analyzed the tumor immunoreactivity against anti-p16 and anti-HPV antibodies as markers of HPV involvement in tumor progression, as well as the correlation with the percentage of intratumoral nuclei immunomarked with the anti-Ki 67 antibody in serial samples. We observed that the percentage of Ki67-positive nuclei increased proportionally with the presence of intratumoral HPV; thus, active HPV infection leads to an increase in the rate of tumor progression. Our results support the implementation of strategies for OPSCC prevention and early diagnosis and can be a starting point for future studies aiming at adapting surgical and oncological treatment according to the HPV stage for better therapeutic results.

## 1. Introduction

A decade ago, the International Agency for Research on Cancer (IARC) established that the high-risk human papillomavirus 16 (HPV16) could be the cause of cellular changes that might lead to the development of oropharyngeal squamous cell carcinoma (OPSCC) [1]. IARC estimates that approximately 29,000 newly diagnosed cases of OPSCC related to HPV infection are recorded annually, representing about 31% of the worldwide number of OPSCC cases [2]. A few meta-analyses on the aspects of HPV involvement in the development of OPSCC showed great geographical heterogeneity, with percentages varying from below 20% in some regions to 24% in Southern Europe and over 60% in North America [3,4]. The clinical, epidemiological, and molecular aspects of OPSCC have also varied depending on the association with other risk factors, such as smoking and alcohol consumption [5].

The number of OPSCC cases connected to the human papillomavirus (HPV) infection, as a carcinogen, recorded a significant increase in the last 20 years, but it was found that infected patients had a higher overall survival rate compared to uninfected ones, regardless of the applied treatment method [6,7,8].

HPV infection with the high-risk strain (HR-HPV) was correlated with the development of OPSCC, and this aspect exceeded the previously established relationship between chronic tobacco use and tumor progression [9,10,11]. Also, the incidence of OPSCC associated with HPV has exceeded the incidence of cervical cancer caused by HPV, and this will continue to evolve even if anti-HPV vaccination rates have begun to increase due to the latent nature of the disease after initial exposure [12].

Studies have shown that the simple detection of HPV in the pathological samples of OPSCC is not enough to support the viral etiology of the cancer since the presence of deoxyribonucleic acid-HPV (HPV-DNA) may reflect a transient relationship or an infection that has not determined tumor progression, rather than an oncogenic transformation caused by HPV [13,14,15]. Also, the immunohistochemical detection of a high cellular level of p16INK4a is the most used technique, but it is not specific to HPV involvement at the tumor level [16,17]. However, it was demonstrated that patients who presented a high expression of p16INK4a but with negative HPV-DNA had a lower survival rate compared to patients who presented a high expression of p16INK4a with positive HPV-DNA [18,19]. Thus, the use of both tests is recommended in order to diagnose tumors caused by an HPV infection [19].

However, the information regarding the accuracy and prognostic value of these biomarker combinations is minimal, and there is a clear necessity to identify the connection between the virus infection and tumor progression for a solid approach to the survival differences of patients with OPSCC, depending on the localization or association with other risk factors [20,21,22].

In an attempt to clarify these aspects, we conducted a study related to OPSCC with the aim of assessing the association between different environmental risk factors and the presence or absence of HPV and p16INK4a in different tumors from this region. Additionally, to highlight the connection, we decided to assess the rate of tumor progression using the Ki67 index (determination of the percentage of Ki67-positive dividing cells in the tumor structure). The advantage of using HPV detection on tissues fixed in paraffin and processed according to the specific immunomarking protocol is that it can analyze a specific area of the tumor structure without using special culture or the polymerization chain reaction (PCR). Thus, the percentage of dividing nuclei immunolabeled with the anti-Ki67 antibody can be determined directly in the positive/negative tumor area for p16/HPV. This study’s main objective is to outline the involvement of all these factors in tumor progression, staging, and post-therapeutic OPSCC progress.

## 2. Materials and Methods

We conducted a retrospective study on 406 out of a total of 6060 patients admitted to the ENT Department within the Emergency County Clinical Hospital of Craiova, Romania which was carried out between 1 January 2018 and 31 December 2022. All patient data were anonymous, and all patients provided their informed consent for the use of their medical data for academic studies. This study was conducted in accordance with the Declaration of Helsinki and approved by the Ethics Committee of the Emergency Clinical County Hospital of Craiova (No. 24378/31.05.2022) and Ethics Committee of the University of Medicine and Pharmacy of Craiova (No. 111/31.05.2022).

We included all patients that clinically presented with an oropharyngeal tumor that had had at least a diagnostic biopsy specimen positive for squamous carcinoma or a premalignant lesion. Lymphomas and other rare histological aspects were excluded.

According to the localization of the studied tumor, all patients were classified into four groups, as follows: Group I included patients with palatine tonsil/tonsillar fossa tumor (TT), Group II included patients with tongue base tumor (ToT), Group III included patients with palatine veil/uvula tumors (PT), and Group IV included patients with tumors with other oropharyngeal localization (OT). The last group mostly included advanced tumors with an unclear anatomic origin, since at the time of diagnosis the tumors were extended in two or more regions.

Each group was subdivided according to gender, rural and urban living environment, to the type of premalignant or malignant lesions, to the status at discharge time (surgically healed, improved or deceased) and to the oncological diagnostic stage (stages I–IV).

After subdividing the groups, using the ME program, we calculated the mean age of the patients and the standard deviation according to the group to which they belonged. We determined the percentages according to patient sex, the number of alcohol or tobacco users and their rural/urban living environment.

Tumor fragments (biopsies or surgically resected specimens) from all 406 patients were sent to the Research Center for Microscopic Morphology and Immunology Craiova, Romania for processing and histopathological analysis.

The tissue fragments were fixed in 10% formalin for a minimum of 7 days and subsequently paraffin embedded according to the pre-established protocols. The obtained tissue blocks were serially sectioned using the HMB350 microtometer equipped with a section transfer in a water bath system (STS microM) at a thickness of 5 μm. The sections obtained were applied on glass slides and Poly-L-Lisine treated slides, subsequently followed by a classical or specific immunohistochemical staining procedure (Table 1).

The slides thus obtained were analyzed with a Nikon Eclipse 44i microscope equipped with a Nikon DS-5Mc 5 megapixel cooled charged-coupled device (CCD) and controlled by the Image-Pro Plus AMS 7 (Media Cybernetics, Rockville, MD, USA) image analysis package.

Immunomarking was performed on serial slides so that the area studied was similar for each type of the analyzed antibody. For anti-HPV and anti-p16 antibodies, the results were expressed as positive marking percentage for each group, while for anti-Ki67 antibody immunostaining, we obtained four images for each sample using a 40× objective, and the total number of Ki67-positive nuclei (automatically calculated by the software) were introduced into the ME program. All the mean values and standard deviations were included in this study.

## 3. Results

### 3.1. Results in Terms of Clinical Aspects and Risk Factors

After dividing the patients into the four groups, we noticed that out of a total of 406 patients, 78.33% belonged to the TT group (tonsillar), 10.84% to the ToT group (tongue), 6.40% to the PT group (palatine) and 4.43% to the OT group (other). Thus, we observed that most tumors were located in the tonsils or the tonsillar fossa, both in men and women. In terms of gender distribution, the comparative study showed that, in the case of TT, 13.84% were women and 86.16% were men, and in the rest of the groups, all patients were men. Regarding their age, we observed that in the group of patients with TT, the mean age of women was equal to 63.05 years old (±10.39 years), and the mean age of men was equal to 62.89 years old (±10.43 years). In ToT, PT and OT groups the calculated mean ages were 63.80 years old (±9.83 years), 63.5 years old (±11.01 years) and 62.06 years old (±10.48 years), respectively. No significant age differences were observed according to the group or the gender of the patients.

Depending on the origin environment, we observed the fact that most patients included in the study were from the urban environment (60.84%) and a smaller percentage were from the rural environment (39.16%). Analyzing the results by gender, the distribution was different since 11.36% of the women were from the rural environment and 88.63% were from the urban environment, while in male patients, the figures were significantly more balanced since 42.54% of cases were from the rural environment and 57.45% were from the urban environment.

Regarding the alcohol consumption, in our study, we observed that 59.09% of women admitted to struggle with chronic alcohol consumption compared to just 16.42% of men. At the same time, 75.69% of men were smokers compared to only 59.09% of women.

All patients included in the study underwent histopathological examination by tumor biopsy. Histopathological examination revealed that 38.68% were benign tumors (most commonly dysplastic papilloma) and 61.32% were malignant tumors. Of the total number of benign tumors, 35.77% were diagnosed in women and 64.23% in men.

According to the clinical diagnostic stage and postsurgical outcomes, malignant tumors from the TT group were distributed as follows:-4.62% Stage I, completely cured after treatment;-5.13% Stage II, completely cured after treatment;-4.1% Stage III completely cured after treatment;-43.59% Stage III improved after treatment;-42.56% Stage IV improved after treatment.

We observed that all cases in the ToT group were diagnosed in Stage IV and had at least a temporary postsurgical improvement. In the case of the PT group, 69.23% improved after the surgery, but 30.77% of the patients encountered severe postsurgical morbidity or died within 30 days after the intervention. Regarding the OT group, all patients were in diagnostic Stage IV and died within 30 days after the intervention (Figure 1).

### 3.2. Results of the Histological and Immunohistochemical Study

During the histopathological study, we analyzed the histological slides obtained after the classical haematoxilin–eosin (HE) staining and special staining with the three previously described antibodies. It was observed that the immunohistochemical reactivity varied according to the type of tumor (benign/malignant) and according to the progression stage of the tumor.

The histopathological and immunohistochemical study of the patients in Group I (TT) resulted in a diagnosis of a percentage of 38.68% premalignant tumors (TT-B), mostly dysplastic papillomas. The samples were analyzed with a 40× objective and a mean total number/field of 146.75 nuclei (±8.55 nuclei), of which a mean number of 50.38 nuclei (±10.44 nuclei) were Ki67 positive/field, representing 34.43% of the total nuclei/field. Also, 65.67% of the total nuclei/field showed a negative immunoreaction for the anti-Ki67 antibody.

From the total of 38.68% of TT-B, 6.29% presented a negative immunoreaction both for the anti-p16 antibody and for the anti-HPV antibody (TT-B-NN), 13.52% presented a negative immunoreaction for the anti-p16 antibody and positive for the anti-HPV antibody (TT-B-NP) and 18.87% showed a positive immunoreaction both for the anti-p16 antibody and for the anti-HPV antibody (TT-B-PP).
The analyzed TT-B-NN had a mean number of nuclei/field 40× of 146.13 nuclei (±9.24 nuclei), of which a percentage of 23.03% of the total nuclei/field was immunomarked, and 77.08% of the nuclei showed a negative reaction for the anti-Ki67 antibody;The analyzed TT-B-NP had a mean number of nuclei/field 40× of 146.03 nuclei (±8.44 nuclei), of which a percentage of 32.31% of the total nuclei/field was immunomarked and 67.84% of the nuclei showed a negative reaction for the anti-Ki67 antibody;The analyzed TT-B-PP had a mean number of nuclei/field 40× of 147.47 nuclei (±8.49 nuclei), of which a percentage of 39.75% of the total nuclei/field was immunomarked, and 60.37% of the nuclei showed a negative reaction for the anti-Ki67 antibody. We thus observed that in cases of premalignant tumors, the percentage of Ki67-positive nuclei increased proportionally with the presence of immunoreactivity against the two anti-p16 and anti-HPV antibodies (Figure 2, Figure 3 and Figure 4).

Also within the TT group, we diagnosed a percentage of 8.49% of Stage I tumors (TT-I), of which 0.63% showed a negative reaction for both the anti-p16 antibody and the anti-HPV antibody (TT-I-NN), 1.26% showed a negative reaction for the anti-p16 antibody and a positive reaction for the anti-HPV antibody (TT-I-NP) and 0.94% showed a positive reaction for both the anti-p16 antibody and the anti-HPV antibody (TT-I-PP). Analyzing the TT-I group with the same 40× objective, we obtained a mean number of 158.22 nuclei/field (±3.36 nuclei), of which 71.42 nuclei (±4.26 nuclei) represented 45.17% nuclei with a positive reaction and 54.86% nuclei with a negative reaction to the anti-Ki67 antibody.

The analyzed TT-I-NN showed a mean number of nuclei/field 40× of 160.50 nuclei (±1.06 nuclei), of which a percentage of 44.70% of the total nuclei/field showed a positive reaction, and 55.30% of the nuclei showed a reaction negative for the anti-Ki67 antibody; The analyzed TT-I-NP showed a mean number of nuclei/field 40× of 156.56 nuclei (±4.55 nuclei), of which a percentage of 44.60% of the total nuclei/field showed a positive reaction, and 55.49% of the nuclei showed a reaction negative for the anti-Ki67 antibody. The analyzed TT-I-PP showed a mean number of nuclei/field 40× of 158.92 nuclei (±1.46 nuclei), of which a percentage of 46.25% of the total nuclei/field showed a positive reaction, and 53.75% of the nuclei showed a negative reaction for the anti-Ki67 antibody. In the case of tumors diagnosed in Stage I, we observed an increase in the number of Ki67-positive nuclei depending on the positivity towards anti-p16 and anti-HPV antibodies, the highest percentage being recorded when both antibodies reacted positively.

From the same group of TT, 3.14% of the cases were Stage II tumors (TT-II), of which 1.26% showed a negative reaction for both the anti-p16 antibody and the anti-HPV antibody (TT-II-NN), 0.94% presented a negative reaction for the anti-p16 antibody and a positive reaction for the anti-HPV antibody (TT-II-NP), 0.31% presented a positive reaction for the anti-p16 antibody and a negative reaction for the anti-HPV antibody (TT-II-PN) and 0.63% showed a positive reaction both for the anti-p16 antibody and for the anti-HPV antibody (TT-II-PP). Analyzing the TT-II batch with the same 40× objective, we obtained an average number of 151.95 nuclei/field (±7.93 nuclei), representing 47.03% nuclei with a positive reaction and 53.06% nuclei with a negative reaction for the anti-Ki67 antibody.

The analyzed TT-II-NN showed a mean number of nuclei/field 40× of 153.69 nuclei (±6.04 nuclei), of which a percentage of 45.25% of the total nuclei/field showed a positive reaction, and 54.78% of the nuclei showed a reaction negative for the anti-Ki67 antibody; TT-II-NP showed a mean number of nuclei/field 40× of 150.33 nuclei (±0.52 nuclei), of which a percentage of 47.29% of the total nuclei/field showed a positive reaction, and 52.72% of the nuclei showed a reaction negative for the anti-Ki67 antibody; TT-II-PN showed a mean number of nuclei/field 40× of 159.00 nuclei, of which a percentage of 47.01% of the total nuclei/field showed a positive reaction, and 52.99% of the nuclei showed a negative reaction for the anti-Ki67; TT-II-PP showed an average number of nuclei/field 40× of 147.38 nuclei (±18.56 nuclei), of which a percentage of 50.20% of the total nuclei/field showed a positive reaction, and 50.04% of the nuclei showed a reaction negative for the anti-Ki67 antibody (Figure 5). Additionally, in the case of tumors diagnosed in Stage II, we observed an increase in the number of Ki67-positive nuclei depending on the positivity to anti-p16 and anti-HPV antibodies, the highest percentage being recorded when both antibodies reacted positively.

A total of 29.25% of the tonsillar tumors (TT) were diagnosed in Stage III (TT-III), of which 2.20% showed a negative reaction for both the anti-p16 antibody and the anti-HPV antibody (TT-III-NN), 1.89% presented a negative reaction for the anti-p16 antibody and a positive reaction for the anti-HPV antibody (TT-III-NP), 0.94% presented a positive reaction for the anti-p16 antibody and a negative reaction for the anti-HPV antibody (TT-III-PN) and 24.21% showed a positive reaction for both the anti-p16 antibody and the anti-HPV antibody (TT-III-PP). Analyzing the TT-III group with the same 40× objective, we obtained a mean number of 148.72 nuclei/field (±7.85 nuclei), representing 54.60% nuclei with a positive reaction and 45.56% nuclei with a negative reaction for the anti-Ki67 antibody.

The analyzed TT-III-NN presented a mean number of nuclei/field 40× of 149.04 nuclei (±10.53 nuclei), of which a percentage of 50.98% of the total nuclei/field showed a positive reaction, and 49.17% of the nuclei showed a reaction negative for the anti-Ki67 antibody; TT-III-NP showed a mean number of nuclei/field 40× of 147.17 nuclei (±6.63 nuclei), of which a percentage of 51.80% of the total nuclei/field showed a positive reaction, and 48.24% of the nuclei showed a negative reaction for the anti-Ki67 antibody; TT-III-PN showed an average number of nuclei/field 40× of 148.50 nuclei (±9.01 nuclei), of which a percentage of 54.96% of the total nuclei/field showed a positive reaction, and 45.29% of the nuclei showed a reaction negative for the anti-Ki67 antibody; TT-III-PP showed an average number of nuclei/field 40× of 148.82 nuclei (±7.79 nuclei), of which a percentage of 55.13% of the total nuclei/field showed a positive reaction, and 45.03% of the nuclei showed a reaction negative for the anti-Ki67 antibody (Figure 5). Also, in the case of tumors from the TT group diagnosed in Stage III, we observed an increase in the number of Ki67-positive nuclei depending on the positivity to anti-p16 and anti-HPV antibodies, the highest percentage being recorded when both antibodies had a positive reaction.

The most advanced tumors in the TT group were diagnosed in Stage IV and represented 26.10% (TT-IV), of which 2.52% showed a negative reaction for both the anti-p16 antibody and the anti-HPV antibody (TT-IV-NN), 1.89% presented a negative reaction for the anti-p16 antibody and a positive reaction for the anti-HPV antibody (TT-IV-NP), 1.26% presented a positive reaction for the anti-p16 antibody and a negative reaction for the anti-HPV antibody (TT-III-PN) and 20.44% showed a positive reaction for both the anti-p16 antibody and the anti-HPV antibody (TT-IV-PP). Analyzing the TT-IV batch with the same 40× objective, we obtained a mean number of 150.02 nuclei/field (±7.28 nuclei), representing 60.29% nuclei with a positive reaction and 26.69% nuclei with a negative reaction for the anti-Ki67 antibody.

The analyzed TT-IV-NN showed a mean number of nuclei/field 40× of 148.31 nuclei (±8.31 nuclei), of which a percentage of 60.29% of the total nuclei/field showed a positive reaction, and 39.91% of the nuclei showed a reaction negative for the anti-Ki67 antibody; TT-IV-NP showed an average number of nuclei/field 40× of 151.38 nuclei (±8.13 nuclei), of which a percentage of 61.20% of the total nuclei/field showed a positive reaction, and 38.98% of the nuclei showed a reaction negative for the anti-Ki67 antibody; TT-IV-PN showed an average number of nuclei/field 40× of 149.13 nuclei (±3.64 nuclei), of which a percentage of 62.88% of the total nuclei/field showed a positive reaction, and 37.13% of the nuclei showed a reaction negative for the anti-Ki67 antibody; TT-IV-PP showed a mean number of nuclei/field 40× of 150.16 nuclei (±7.34 nuclei), of which a percentage of 76.85% of the total nuclei/field showed a positive reaction, and 23.31% of the nuclei showed a reaction negative for the anti-Ki67 antibody (Figure 5). Additionally, in the case of tumors from the TT group diagnosed in Stage IV, we observed an increase in the number of Ki67-positive nuclei depending on the positivity towards anti-p16 and anti-HPV antibodies, the highest percentage being recorded when both antibodies had a positive reaction.

Regarding group II of patients diagnosed with ToT, it was observed that the tumors were advanced, all in Stage IV (100%), of which 11.36% showed a positive reaction for the anti-p16 antibody and a negative reaction for the anti-HPV (TT-IV-PN) and 88.64% showed a positive reaction both for the anti-p16 antibody and for the anti-HPV antibody (TT-IV-PP). Analyzing the ToT-IV group with the same 40× objective, we obtained a mean number of 153.14 nuclei/field (±10.94 nuclei), representing 78.23% nuclei with a positive reaction and 22.06% nuclei with a negative reaction for the anti-Ki67 antibody.

The analyzed ToT-IV-PN presented a mean number of nuclei/field 40× of 171.25 nuclei (±19.67 nuclei), of which a percentage of 72.02% of the total nuclei/field showed a positive reaction, and 28.41% of the nuclei showed a reaction negative for the anti-Ki67 antibody. The analyzed ToT-IV-PP showed a mean number of nuclei/field 40× of 150.82 nuclei (±6.78 nuclei), of which a percentage of 79.02% of the total nuclei/field showed a positive reaction, and 21.14% of the nuclei showed a reaction negative for the anti-Ki67 antibody (Figure 3, Figure 4 and Figure 5). We observed that in the case of ToT, the percentage of Ki67-positive nuclei increased proportionally to the positivity towards anti-p16 and anti-HPV antibodies and that their percentage was higher compared to TT-IV, registering a difference of 17.94% in favor of ToT-IV.

Regarding Group III of patients diagnosed with PT, we observed that all the tumors were advanced, all in Stage IV (PT-IV) (100%), of which 19.23% showed a positive reaction for the anti-p16 antibody and negative reaction for the anti-HPV antibody (PT-IV-PN) and 80.77% showed a positive reaction for both the anti-p16 antibody and the anti-HPV antibody (PT-IV-PP). Analyzing the PT-IV batch with the same 40× objective, we obtained an average number of 149.29 nuclei/field (±7.70 nuclei), representing 82.55% of nuclei with a positive reaction and 17.61% of nuclei with a negative reaction for the anti-Ki67 antibody.

The analyzed PT-IV-PN showed a mean number of nuclei/field 40× of 146.80 nuclei (±6.66 nuclei), of which a percentage of 81.09% of the total nuclei/field showed a positive reaction, and 18.97% of the nuclei showed a reaction negative for the anti-Ki67 antibody; PT-IV-PP showed a mean number of nuclei/field 40× of 149.88 nuclei (±7.95 nuclei), of which a percentage of 82.90% of the total nuclei/field showed a positive reaction, and 17.29% of the nuclei showed a reaction negative for the anti-Ki67 antibody (Figure 5). We observed that in the case of PT, the percentage of Ki67-positive nuclei was increased proportionally with the positivity to anti-p16 and anti-HPV antibodies and that their percentage was higher compared to TT-IV and ToT-IV, differences of 22.26% and, respectively, 4.32% in favor of PT-IV.

Regarding Group IV of patients diagnosed with OT, we observed that all tumors were advanced in Stage IV (OT-IV) (100%), of which 22.22% showed a positive reaction for the anti-p16 antibody and a negative reaction for the anti-HPV antibody (OT-IV-PN) and 77.78% showed a positive reaction for both the anti-p16 antibody and the anti-HPV antibody (PT-IV-PP). Analyzing the OT-IV group with the same 40× objective, we obtained a mean number of 150.85 nuclei/field (±7.62 nuclei), representing 83.29% nuclei with a positive reaction and 16.78% nuclei with a negative reaction for the anti-Ki67 antibody.

The analyzed OT-IV-PN showed a mean number of nuclei/field 40× of 152.81 nuclei (±7.62 nuclei), of which a percentage of 79.07% of the total nuclei/field showed a positive reaction, and 20.93% of the nuclei showed a reaction negative for the anti-Ki67 antibody; OT-IV-PP presented a mean number of nuclei/field 40× of 150.29 nuclei (±7.81 nuclei), of which a percentage of 84.49% of the total nuclei/field showed a positive reaction, and 15.51% of the nuclei showed a reaction negative for the anti-Ki67 antibody (Figure 5). It was observed that in the case of OT, the percentage of Ki67-positive nuclei was increased proportionally to the positivity to anti-p16 and anti-HPV antibodies and that the percentage of positive nuclei was higher compared to those of TT-IV, ToT-IV, PT-IV records; there are differences of 23%, 5.06% and, respectively, 0.74% in favor of OT-IV (Figure 5).

## 4. Discussion

OPSCC is the sixth most common cancer in the world [23]. Most of the time, associated risk factors lead to tumor progression, but malignant transformation can be prevented by the fact that both smoking and alcohol, which were considered major risk factors, can be avoided [24]. Their synergic effect accelerates the process of tumor transformation of premalignant lesions in the oropharynx [25].

In 2007, the IACR concluded that there is sufficient evidence to prove that cigarette smoke has a carcinogenic effect and can cause oral cancer [1], and the risk of developing such cancer is three times higher in smokers in comparison with non-smokers [26]. It has also been shown that the risk drops by 35% in people who have stopped smoking for at least four years compared to those who continue to smoke; the risk does not differ in people who have not smoked for more than 20 years compared to people who have never smoked [27,28,29,30]. Similar to these data, in our study, we observed that approximately two thirds of the patients were chronic smokers (73.89%) and developed OPSCC.

Another risk factor associated with cellular changes that predispose to epithelial tumor transformation is the consumption of alcohol (ethanol), which acts both locally and systemically. It increases the permeability of the oropharyngeal mucosa, dissolves epithelial lipid components, induces epithelial atrophy, disrupts normal DNA synthesis and repair, causing genotoxicity and mutagenicity. It also induces a decrease in salivary flow, affects liver function in terms of its action against toxic or potentially carcinogenic compounds, affecting the body’s innate and acquired defense functions, predisposing the body to infections and the malignant transformation of some tissue structures [31].

In our study, we observed that women consuming alcohol were three times more susceptible to the occurrence of TT compared to men. This can also be explained by the fact that women can present low immunity in certain periods of life, post-partum or menopause due to the decrease in the amount of estrogen hormone, and this is in opposition to some studies that claimed that men are more susceptible to bacterial, fungal and viral infections due to the fact that they have a single X chromosome, correlated with various mutations that induce immunodeficiency [32,33]. Globally, in this study, there were no significant differences in terms of the age and gender of the patients in the analyzed groups, but we noticed that the male gender was more prone to the occurrence of TT.

Regarding the origin of the patients included in the study, we noticed that more than half of them were from the urban environment, but the studies showed that the survival rate of these patients is better compared to the patients from the rural environment [34]. The lower rate of patients from rural areas can also be attributed to the fact that many patients do not have access to medical services, do not have access to medical information or cannot afford to go to the urban area for additional medical investigations. This fact also contributes to a better survival of urban patients in not very advanced stages through easier access to medical services [35].

The most recent major risk factor associated with OPSCC is the HPV infection since viral DNA and viral proteins were demonstrated in various proportions in cancerous cells. However, the intimate role of viral infection is not fully understood since the infection was proven also in normal mucosa and precancerous lesions. Another question that still poses problems is whether the infection occurs and persists after neoplastic transformation, at least in a still unknown proportion of the cases due to loss of defense and repair mechanisms.

In our study, we observed an increasing positivity both for anti-p16 and anti-HPV antibody immunmarking from Stage I to more advanced stages of OPSCC carcinomas which can be interpreted in a few ways. Either the infection is present but the proteic expression becomes stronger with the more advanced stage due to successive acquiring of replication errors, or the infection takes place after neoplastic transformation, being responsible for tumoral boost. We also noticed a greater positivity proportion for anti-p16 and anti-HPV in premalignant lesions, greater than that for OPSCC Stages I and II, meaning that the HPV infection is present and has molecular expression from the very beginning of the neoplastic transformation, at least for a variable proportion of the cases (calculated around 47% for our region) and has the potential to initiate the neoplastic sequence.

We also observed that the percentage of nuclei immunomarked with the anti-Ki67 antibody increased proportionally with the positivity of the immunoreactions with the other two antibodies used, anti-p16 and anti-HPV. Ki67-positive nuclei are associated with tumor growth and proliferation; these nuclei are prognostic and predictive indicators in the evaluation of patients diagnosed with different types of cancer, such as the oropharyngeal one in our case [36].

HPV-induced changes in the oropharyngeal mucosa have been associated with the development of malignant lesions, but studies have shown that the survival of these patients was improved and they responded better to treatment [36]. However, the reasoning that leads to this behavior is for the moment poorly understood, which is why we wanted to develop this idea. We noticed that HPV-positive patients in whom the tumor suppressor protein p16INK4a was also positive were associated with increased tumor proliferation. Tumor suppressor protein p16INK4a has the role of slowing cell division from the G1 phase of the cell cycle to the S phase, thus acting as a tumor suppressor. When the gene encoding this protein, CDKN2A, undergoes mutations and/or deletions, the antitumor action is ineffective or non-functional, thus accelerating the cell cycle and being correlated with tumor proliferation [37,38]. Our results tend to demonstrate that induced HPV or positive HPV OPSCC have a higher proliferation rate and, theoretically, are more aggressive. Independently of their localization, we found the greatest rate of positivity (ranging between 70% and 80%) both for anti-HPV and anti-p16 in Stage IV carcinomas. But the same rapid growth rate can be the explanation for a better response to the oncologic treatment, since dividing neoplastic cells are more susceptible to chemotherapy and especially radiotherapy. Surgical therapy, as expected, may have worse outcomes since it does not interfere with the intimate process of cell proliferation. This observation is congruent with other studies that reported an advanced stage at diagnosis for HPV-positive OPSCC, but better responses to chemoradiation when compared with HPV-negative cancers while surgery did not bring any advantages [39].

The expression of p16INK4a represents a prognostic biomarker especially in cancers caused by HPV, which a strong indicator of the progression of the disease after primary surgical treatment [40]. Its presence shows the possibility of a better prognosis when the tumor lesion is diagnosed [41,42,43]. In our study, we observed that there was a directly proportional correlation between the number of Ki67-positive nuclei and the positivity towards anti-p16 and anti-HPV antibodies, the highest percentage, regardless of stage, recorded when both antibodies reacted positively. Thus, we support the hypothesis that the CDKN2A gene mutation contributes to uncontrolled tumor proliferation [37,38]. Also, in the case of the TT, TT-IV, ToT-IV groups, we obtained the same directly proportional correlation of the number of Ki67-positive nuclei and the positivity towards anti-p16 and anti-HPV antibodies, the highest percentage being recorded when both antibodies reacted positively.

Molecular, polymerase chain reaction (PCR) studies have shown that HPV integration into the genome has been frequently observed in cervical carcinomas and OPSCC [44,45] and that recent research can analyze HPV copy number and accurately detect epizomal and integrated forms in both cell lines and tissues grown in organotypic raft culture. Our study did not aim at comparing histopathology with PCR, as both methods have different advantages; thus, the histopathological method specifically analyzes a tumor area, serially sectioned, immunolabeled with the three anti-p16/HPV/Ki67 antibodies studied, thus being able to perform descriptive statistical studies, whereas molecular methods can detail whether the genome is episomal or integrated. All these aspects suggest that deletion, hypermethylation, mutation, or overexpression of p16 is associated with tumor proliferation of OPSCC, and HPV is a cause of the appearance of cellular changes that led to malignant degeneration [37,38].

## 5. Conclusions

Chronic smoking, alcohol consumption or the origin urban environment can contribute to the onset of oropharyngeal tumors, but the presence of HPV was more frequently associated with malignant tumor transformation and diagnosis in advanced stages. Further studies are needed to clarify the role of HPV in tumor onset, tumor progression or boost since the risk factors are intricate in most of the cases.

HPV positivity markers had a maximum expression rate in advanced stages pleading for high growth rate as proved by the Ki67 index that increased proportionally to the anti-HPV and anti-p16 immunomarking.

The presence of HPV is a prognostic factor for overall survival and a predictive marker of response to post-interventional treatment. Also, the immunoreaction for p16 can be used as a surrogate marker for HPV involvement on exposed tissues. Combining anti-HPV, anti-p16 and Ki67 immunomarking in the primary diagnosis of OPSCC might be an option for choosing the best approach therapy that seems to exclude extensive surgery at least for advanced cancers.

The findings regarding the association of HPV with the changes in the gene that encodes p16INK4a and the induction of tumor proliferation can be the basis of future studies and the choice of therapeutic strategies in the case of oropharyngeal tumors, potentially leading to the accuracy of the diagnosis and pre/post-therapeutic prognostic values related to the presence of HPV.

## Figures and Tables

**Figure 1 pathogens-12-00984-f001:**
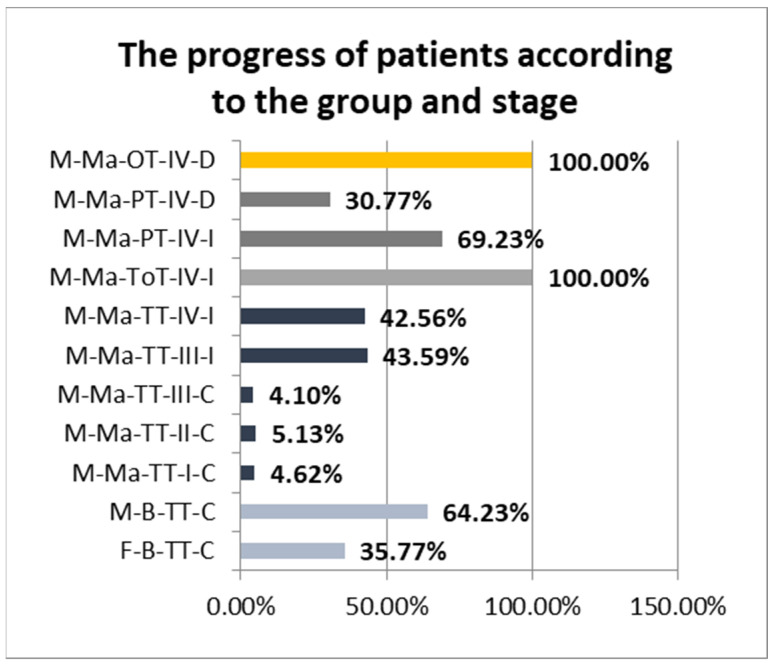
The progress of patients according to the group and stage (F: female; M: male; TT: palatine tonsil or tonsillar fossa tumor; ToT: tongue base tumor; PT: palatine veil or uvula; OT: tumors with other oropharyngeal locations; B: benign; Ma: malign; I, II, III, IV–staging; I: improved; C: cured; D: poor health state or death).

**Figure 2 pathogens-12-00984-f002:**
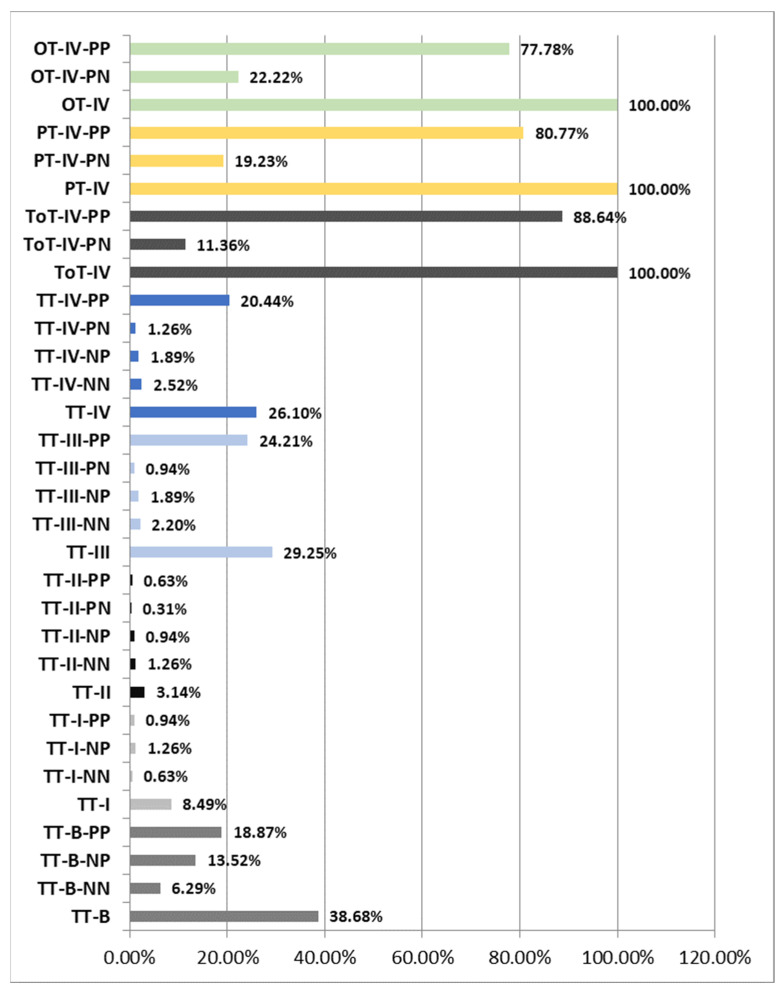
Percentage distribution of tumors in the 4 study groups according to the reactivity to anti-p16 and anti-HPV antibodies. TT: palatine tonsil or tonsillar fossa tumor; ToT: tongue base tumor; PT: palatine veil or uvula; OT: tumors with other oropharyngeal locations; I, II, III, IV–stages; NN: negative reaction for both the anti-p16 antibody and the anti-HPV antibody; NP: negative reaction for the anti-p16 antibody and positive for the anti-HPV antibody; PN: positive reaction for the anti-p16 antibody and negative for the anti-HPV antibody; PP: positive reaction for both the anti-p16 antibody and the anti-HPV antibody.

**Figure 3 pathogens-12-00984-f003:**
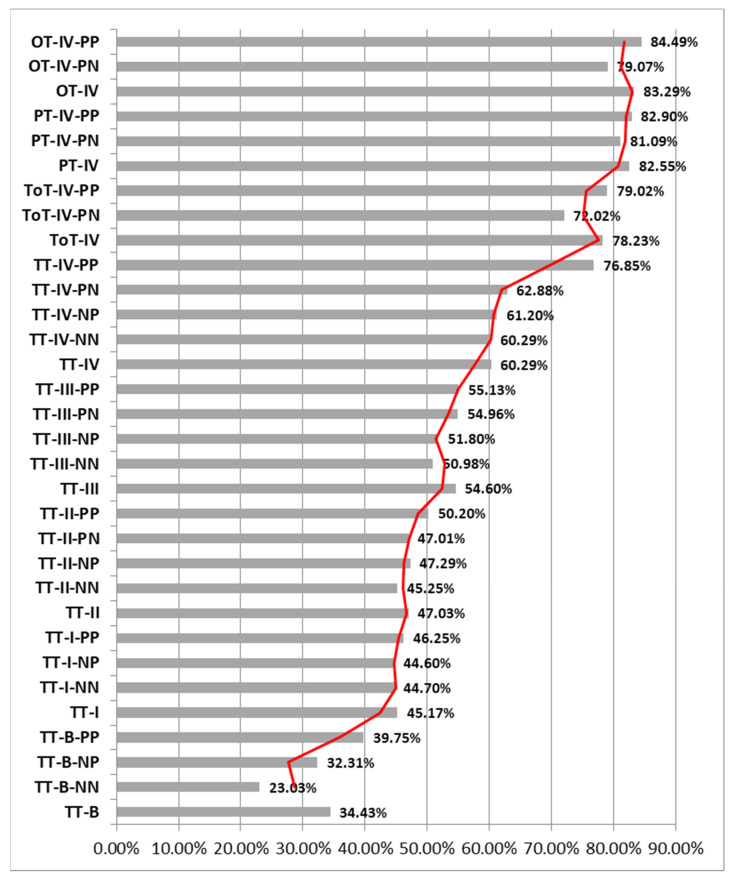
Percentage distribution of Ki67-positive nuclei according to the type and stage of tumors in the 4 groups. A gradual increase in the percentage of Ki67-positive nuclei directly proportional to the increase in tumor stage is observed (red line). TT: palatine tonsil or tonsillar fossa tumor; ToT: tongue base tumor; PT: palatine veil or uvula; OT: tumors with other oropharyngeal locations; B: benign; I, II, III, IV–staging; NN: negative reaction for both the anti-p16 antibody and the anti-HPV antibody; NP: negative reaction for the anti-p16 antibody and positive for the anti-HPV antibody; PN: positive reaction for the anti-p16 antibody and negative for the anti-HPV antibody; PP: positive reaction for both the anti-p16 antibody and the anti-HPV antibody.

**Figure 4 pathogens-12-00984-f004:**
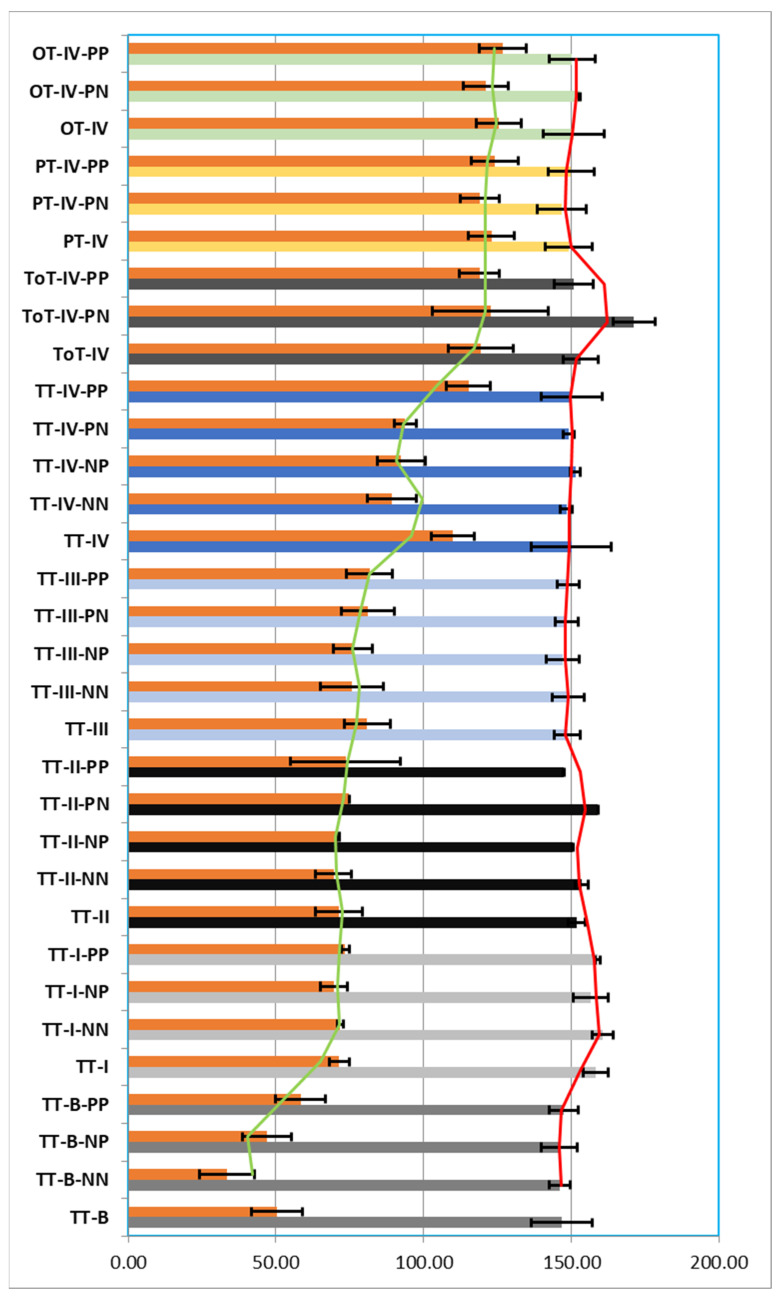
Distribution of the total number of nuclei/field 40× and of the total number of nuclei immunmarked with the anti-Ki67 antibody/field 40×, depending on the type and stage of the tumors in the 4 groups. We observe a gradual increase in the number of Ki67-positive nuclei, directly proportional to the tumor stage (green line), while the total number of nuclei/field 40× remained quasi-constant (red line). TT: palatine tonsil or tonsillar fossa tumor; ToT: tongue base tumor; PT: palatine veil or uvula; OT: tumors with other oropharyngeal locations; B: benign; I, II, III, IV–stages; NN: negative reaction for both the anti-p16 antibody and the anti-HPV antibody; NP: negative reaction for the anti-p16 antibody and positive for the anti-HPV antibody; PN: positive reaction for the anti-p16 antibody and negative for the anti-HPV antibody; PP: positive reaction for both the anti-p16 antibody and the anti-HPV antibody.

**Figure 5 pathogens-12-00984-f005:**
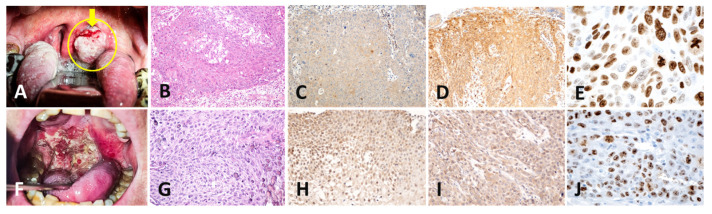
(**A**–**J**)—Malignant tumor from the TT group. (**A**): Macroscopic image showing an infiltrative-vegetative tumor at the level of the left palatine tonsil (arrow and yellow border); (**B**): microscopic aspect of the tonsilar tumor in classical HE staining, 10×; (**C**): positive immunoreaction to the anti-p16 antibody 10×; (**D**): positive immunoreaction to the anti-HPV antibody 10×; (**E**): immunoreaction against the anti-Ki67 antibody of the tonsillar tumor area; the nuclei in the division phase can be identified, stained with brown, 40×; (**F**–**J**): malignant tumor from the OT group. (**F**): macroscopic image showing the presence of the tumor destroying the oropharynx (arrow and yellow border); (**G**): microscopic aspect of the oropharyngeal tumor in classical HE staining, 10×; (**H**): positive immunoreaction to the anti-p16 antibody of the oropharyngeal tumor area, 10×; (**I**): positive immunoreaction to the anti-HPV antibody of the oropharyngeal tumor area; (**J**): positive immunoreaction to the anti-Ki67 antibody of the oropharyngeal tumor area; we can identify the nuclei in the division stained with brown, 40×; TT: tongue base/tonsillar fossa tumor; OT: tumors with other oropharyngeal locations; HE: hematoxylin and eosin staining; HPV: human papillomavirus.

**Table 1 pathogens-12-00984-t001:** The primary antibody.

Antibody	Manufacturer	Clone	Antigenic Exposure	Secondary Antibody	Dilution	Labeling
Anti-Ki67	Dako	MIB-1	EDTA, pH 9	Monoclonal mouse anti-human Ki67	1:50	Cells in division in the G1, S, G2 and M phase
Anti-p16	Invitrogen	1D7D2	Citrate, pH 6	p16INK4a antibody (1D7D2)	1:1000	Tumor suppressor protein
Anti-HPV	Abcam	K1H8	Citrate, pH 6	Mouse monoclonal (K1H8) to HPV	1:50	HPV-infected cells

EDTA: Ethylenediaminetetraacetic acid; IHC: Immunohistochemical; Ki67: Marker of proliferation; HPV: Human papillomavirus.

## Data Availability

Data was collected from the Otorhinolaryngology Clinic of the Emergency County Clinical Hospital of Craiova, Romania This data is not available online but can be provided upon formal request.

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
