# Peer review of "HPV and Other Risk Factors Involved in Pharyngeal Neoplasm—Clinical and Morphopathological Correlations in the Southwestern Region of Romania"

_pathogens, 2023, doi:10.3390/pathogens12080984_

Round 1

Reviewer 1 Report

The study does not bring anything new; it describes results obtained and presented in a much better form in many previous studies.

 The abstract is unclear on lines 33-34 "In an immunohistochemical study, we analyzed the contribution of anti-p16 and anti-HPV antibody immunoreactivity as markers of HPV involvement in tumor development, as well as Ki 67 expression." "What did the authors detect in the FFPE samples? The same error appears throughout the text.

  The presentation of the results is very messy and unclear. Given the small number of results, it is unnecessarily long.

 The authors make it look as if HPV has recently been added to the list of OPSCC risk factors; however, the 8th edition of the TNM clinical cancer classification already introduces p16 positive oropharyngeal tumors into clinical practice.

  There is a complete lack of statistics in the text and therefore the authors have no support in any of their results presented in manuscript

 I do not consider myself an expert in English, however the quality of this text is very poor, due to this there are errors somewhere.

Author Response

In the PDF document you have attached the answers. Thank you! 

Reviewer 2 Report

The submitted work is a retrospective study on 406 patients diagnosed with OPSCC in Southwest Romania. Patients were classified according to tumoral type, location and diagnostic stage, demographic characteristic, risk factors, histologic and immunohistochemical features. Most of the patients were male from urban areas with smoking habit, while most of the female patients were diagnosed with tonsillar OPSCC and had a history of chronic alcohol abuse. Using immunohistochemistry the proportion of immunoreactivity to anti-p16 and anti-HPV antibodies as markers of HPV implication in tumoral development were explored, completed by the expression of Ki 67. The percentage of Ki67 positive nuclei increased proportionally with the presence of HPV (p16) immunoreactivity.

Two important points makes this study particularly interesting: analysis in Southwest Romania with well-discussed results, and the correlation between the intensity of p16/HPV immunohistochemistry and the proliferation rate (Ki-67-positivity) in tumor.

Comments

General

Do authors see it more appropriate, or comparable appropriate, the use of the Abcam ab75574 antibody, which delivers in situ/local information for HPV detection, than the HPV PCR, which is a standard method for HPV detection in OPSCC?

Detailed

Methods

Could you mention an ethic approval and patients consent in the Methods part?

Table 1:

MIB1 is a clone, correct but MA5- 17054 and ab75574 are catalogue numbers.

Author Response

(The authors gave the same response as above.)

Reviewer 3 Report

In this study, Mogoanta et al. carry out analysis of pharyngeal neoplasm biopsies with an aim to identify correlations in tumour type, stage and demographic with risk factors and immunohistological feature, with an aim to identify prognostic markers and potential observations that would enhance treatment triaging.

The results are comprehensively described, and ultimately are consistent with the findings of other studies. This is a descriptive study which bolsters existing studies that ultimately suggests the co-detection of p16, Ki67 and HPV in OPSCC in determining cancer stage and therapeutic options.

I have no major criticisms of this study. Not all OPSCC HPV contain integrated HPV DNA into the human genome, so it would be interesting to look at the status of HPV in the samples - whether the genome is episomal or integrated, and whether that status correlates with prognosis. Perhaps this could be included in future directions.

Author Response

(The authors gave the same response as above.)

Round 2

Reviewer 1 Report

The authors have made only minor changes to the manuscript. In my opinion, the main problems remain: the paper is unnecessarily large and unclear, and the results and conclusions presented are not supported by statistical analyses